# Hydrophobization of Tobacco Mosaic Virus to Control the Mineralization of Organic Templates

**DOI:** 10.3390/nano9050800

**Published:** 2019-05-24

**Authors:** Petia Atanasova, Vladimir Atanasov, Lisa Wittum, Alexander Southan, Eunjin Choi, Christina Wege, Jochen Kerres, Sabine Eiben, Joachim Bill

**Affiliations:** 1Institute for Materials Science, University of Stuttgart, Heisenbergstr. 3, 70569 Stuttgart, Germany; choi@is.mpg.de (E.C.); bill@imw.uni-stuttgart.de (J.B.); 2Institute of Chemical Process Engineering, University of Stuttgart, Böblinger Straße 78, 70199 Stuttgart, Germany; vladimir.atanasov@icvt.uni-stuttgart.de (V.A.); jochen.kerres@icvt.uni-stuttgart.de (J.K.); 3Institute of Biomaterials and Biological Systems, University of Stuttgart, Pfaffenwaldring 57, 70569 Stuttgart, Germany; lisawittum1@yahoo.de (L.W.); christina.wege@bio.uni-stuttgart.de (C.W.); sabine.eiben@gmx.de (S.E.); 4Institute of Interfacial Process Engineering and Plasma Technology, University of Stuttgart, Pfaffenwaldring 31, 70569 Stuttgart, Germany; alexander.southan@igvp.uni-stuttgart.de

**Keywords:** tobacco mosaic virus, ZnS, bio/inorganic hybrid materials, hydrophobization, polymer coupling

## Abstract

The robust, anisotropic tobacco mosaic virus (TMV) provides a monodisperse particle size and defined surface chemistry. Owing to these properties, it became an excellent bio-template for the synthesis of diverse nanostructured organic/inorganic functional materials. For selective mineralization of the bio-template, specific functional groups were introduced by means of different genetically encoded amino acids or peptide sequences into the polar virus surface. An alternative approach for TMV surface functionalization is chemical coupling of organic molecules. To achieve mineralization control in this work, we developed a synthetic strategy to manipulate the surface hydrophilicity of the virus through covalent coupling of polymer molecules. Three different types of polymers, namely the perfluorinated (poly(pentafluorostyrene) (PFS)), the thermo-responsive poly(propylene glycol) acrylate (PPGA), and the block-copolymer polyethylene-block-poly(ethylene glycol) were examined. We have demonstrated that covalent attachment of hydrophobic polymer molecules with proper features retains the integrity of the virus structure. In addition, it was found that the degree of the virus hydrophobicity, examined via a ZnS mineralization test, could be tuned by the polymer properties.

## 1. Introduction

In the process of fast-growing development of new nanostructured functional materials, a huge variety of organic materials, in particular biological objects, have been used to synthesize organic/inorganic hybrids with desired properties for nanotechnological applications [1,2,3,4]. Among them, tobacco mosaic virus (TMV) is a robust tube-like plant virus, harmless for humans and animals and can be produced in scalable amounts in a green house. It has been utilized intensively as a bio-template due to its anisotropic structure, high uniformity in size and shape and defined surface chemistry. Applying ‘bottom-up‘ approaches such as evaporative self-assembly of the capsids by convective assembly and controlled evaporation methods, homogeneous virus monolayers, aligned TMV stripes and nanowires have been produced [5,6,7,8]. TMV has been used as a template to deposit various inorganic materials (metals and semiconductors) on the exterior or the interior surface of the capsid [9,10,11]. The virus itself and TMV-based hybrids have found application in construction of functional devices like field-effect transistors (FETs) [12] and sensors [13,14,15,16], in Li-ion battery [17], etc. Electroconductive materials have been generated making use of the electrostatic interactions between TMV and the polymers polyaniline and polypyrrole, which enabled in-situ polymerization of the polymers on the TMV surface [18]. Most of these studies have been performed using either wild type (wt) TMV, cysteine mutants within the first 3 amino acids of the coat protein (CP) or a C-terminal lysine mutant. An additional number of TMV CP mutants exist [19,20,21], however especially those changing the surface charge interfere with the natural assembly process [22], forming nanotubes without incorporation of stabilizing RNA. These nanotubes vary greatly in length and are often liable to disassembly upon small changes in buffer conditions making them unavailable for mineralization studies. Although there has been a quite successful attempt to stabilize these RNA-free virus-like particles by introducing an inter CP disulfide bridge in the inner channel of the nanotubes and thus opening up the possibility to use bacterially expressed CP mutants for templating [23,24].

Another strategy to increase the possible surface properties of TMV is to use the existing stable mutants for chemical modification by coupling, which has been done to add mineralization inducing peptides, chemicals for magnetic resonance imaging, whole enzymes and polymers [25]. Recently, a DNA-controlled “stop-and-go” strategy was established to assemble two distinct, selectively addressable CPs variants with RNA into artificial TMV nanotubes with highly defined longitudinal subdomains [26]. The presence of two domains consisting of CPs with different functional groups on one artificial TMV nanoparticle [27] gives the opportunity of using, e.g., “click reactions” to specifically couple organic molecules to only one part of the particle and thereby to synthesize Janus-type TMV particles with a hydrophilic and a hydrophobic portion. The amphiphilicity of classical Janus particles provides unique chemical and physical properties, not accessible for their homogeneous counterparts. Owing to their amphiphilic, magnetic, catalytic or optical properties, they have found numerous applications in fields like drug delivery and catalysis, as surfactants and building blocks for complex 3D nanostructures, for water-repellent coatings, etc [28]. Furthermore, such particles provide additional perspectives towards exploring the influence of the genetic modification of TMV CPs on the nucleation and growth of inorganic materials within one artificial TMV particle, i.e., on the guidance of mineralization reactions. While TMV CPs can be engineered genetically or modified chemically to introduce different hydrophilic functionalities on the virus surface, TMV hydrophobization is a challenging task. An important requirement by choosing viable reaction conditions is the prevention of the hydrophilic complex protein structure from disassembly in the presence of a hydrophobic surrounding. In this regard, A. J. Patil et al. have modified the surface of cowpea mosaic virus (CPMV) with the anionic polymer-surfactant poly(ethylene glycol) 4-nonylphenyl 3-sulfopropyl ether through electrostatically directed assembly [29]. However, electrostatic interactions do not allow control over a local specific functionalization of the virus surface, which in contrast can be achieved via covalent coupling of organic molecules. Another important point concerns the degree of virus hydrophobization required to suppress subsequent mineralization, if partially coated particles are sought after. Therefore, in this work we used wt-TMV and a thiol-displaying TMV-Cys mutant [27] to examine different synthesis strategies and to establish an approach for covalent coupling of polymer molecules to the virus surface, with the aim to produce stably functionalized viruses with sufficient hydrophobicity to suppress mineral deposition on the viral protein coat. For the bioconjugation reactions, the highly hydrophobic perfluorinated (poly(pentafluorostyrene) (PFS)), the thermo-responsive poly(propylene glycol) acrylate (PPGA), which becomes hydrophobic by increasing the temperature and the block-copolymer polyethylene-block-poly(ethylene glycol) ((PE)-b-(PEG)) with a well distinguished hydrophobic part were used. We demonstrate that the integrity of the virus particles can be preserved after covalent attachment of hydrophobic polymer molecules, compatible with the structure of the virus. The achieved hydrophobicity was verified via mineralization reaction applied to the modified virus surface with ZnS, in order to examine if the degree of virus hydrophobicity can be controlled by the choice of the conjugated polymer.

## 2. Materials and Methods

Materials Disodium hydrogen phosphate, acetone, sodium dodecyl sulfate, tris(hydroxymethyl)aminomethane, acetic acid, sodium carbonate, ammonium persulfate, tetramethylenediamine, sodium chloride and sodium thiosulfate were obtained from Carl Roth (Karlsruhe, Germany). Triphenylphosphine, hydrazine hydrate, PE-b-PEG-OH, N-hydroxyphthalimide (97%), sodium nitrite (97%, ACS), p-toluenesulfonic acid monohydrate (ACS), p-aminoacetophenone (0.15 M in acetonitrile), diisopropyl azodicarboxylate (95%), zinc chloride (≥98%) and sodium sulfide nonahydrate (≥98%) were obtained from Sigma-Aldrich. Poly(propylene glycol) acrylate (*M_n_* = 475 g mol^−1^), silver nitrate, formaldehyde, Coomassie-Brilliant-Blue R250 and dihydrogen phosphate were obtained from Merck (Darmstadt, Germany). Acrylamide/Bis Solution, 37.5:1 was purchased from Serva (Heidelberg, Germany). Pentaflourostyrene 98% was purchased from ABCR GmbH. All solvents used in this study were in HPLC grade and used as received.

Substrate cleaning: Silicon wafers (100, p-doped polished wafers, Silchem, Germany) were used as substrates. To get a hydrophilic surface, they were thoroughly cleaned prior to use applying the following procedure: 10 min sonication in ultrapure water, 10 min sonication in ethanol/acetone (1:1, v/v), 10 min O_2_ plasma treatment (30 W) and 10 min sonication in ultrapure water. After each sonication step, the substrates were washed 10 times with the corresponding solvent and dried under Ar stream. Silicon substrates with reduced hydrophilicity (not plasma treated) were cleaned only successively with ultrapure water, ethanol and acetone.

Preparation and purification of TMV: wt-TMV of the type strain U1 as well as the TMV-Cys mutant [27] were propagated in Nicotiana tabacum ‘Samsun’ nn plants and purified by PEG precipitation as described before [30]. The virus was stored in 10 mM sodium potassium phosphate (SPP) buffer at pH 7.4 at 4°C. Buffer-free virus solutions were obtained by resuspension of TMV in ultrapure water after pelleting the virus in a Beckman ultracentrifuge at 35,000 rpm (corresponding to an average g force of 95,800) for 105 min at 4 °C using a 45 Ti rotor and resuspension in ultrapure water.

TMV immobilization: A droplet (3 µL, 0.2 mg mL^−1^) buffer-free virus solution was spotted onto a substrate surface and incubated for 10 min in a closed chamber. Then, the droplet was removed and the substrate with the immobilized viruses was dried under argon stream.

Synthesis of poly(pentafluorostyrene)(PFS): The synthesis of PFS has already been reported elsewhere [31]. Briefly, emulsion polymerization reaction of pentafluorostyrene in conditions similar to styrene polymerization was used. The molecular weight was optimized to obtain high values (M_n_ = 52 kDa, M_w_ = 124 kDa and MWD = 2.4), but allowing good solubility of the polymer in solvents like tetrahydrofuran (THF) and CHCl_3_.

Click reaction of TMV-Cys with PFS: Due to the limited solubility of PFS in N,N-dimethylacetamide (DMAc), first 1 wt% PFS in DMAc was prepared at 90 °C for 24 h. The solution was filtered through 0.2 µm filter, and silicon substrates with immobilized viruses were let to react with the filtered polymer solution for 30 min at room temperature. After the treatment, the substrates were washed with DMAc and dried. For reaction in tetrahydrofuran (THF), silicon substrates with immobilized TMV-Cys particles were placed in a vessel with 1 wt% PFS in THF/trimethylamine (TEA) solution at room temperature for 2 h. Then, the substrates were thoroughly washed with THF and dried.

Synthesis of TMV-Cys-poly(propylene glycol) acrylate (PPGA): All solutions were stored in an ice-bath for at least 30 min to prevent warming up of the PPGA. For coupling, 1–4 mg/mL of the virus solutions in 10 mM SPP buffer were incubated with the same volume of PPGA (M_n_ = 475) diluted in 10 mM SPP buffer to give a molecular ratio from 1:1000 to 1:10,000 of CP to PPGA. The solution was incubated in a cooled shaker at 4 °C and 300 rpm for 72 h. Before loading of the functionalized TMV-Cys on a 16/60 Sephacryl S-500 column, bulk PPGA was removed by precipitation of the virus by ultracentrifugation at 34,000 rpm for 2 h at 4°C in a pre-cooled 45 Ti rotor. As sample and running buffer 50 mM SPP containing 150 mM NaCl was used. Purification was performed on an ÄKTApurifier system in a cooling cabinet at a flowrate of 0.5 mL/min, and 1 mL fractions were collected after 75 min until the end of the purification at 250 min. The fractions containing TMV were determined by UV-Vis analysis. Fractions showing an absorbance ratio of 260 nm and 280 nm of 1.19 were pooled and concentrated using an Amicon^®^ Ultra 4 mL centrifugal filter unit with 50,000 nominal molecular weight limit (NMWL) cut-off (Merck, Darmstadt, Germany).

Synthesis of PE-b-PEG phthalimide **(2)**: A three-neck round flask was flame-dried under nitrogen stream. Triphenylphosphine (PPh_3_, 1.377 g, 5.25 mmol), N-hydroxyphthalimide (0.895 g, 5.5 mmol), PE-b-PEG-OH (**1**) (11.25 g, 5 mmol, Mw = 2250 g/mol) and anhydrous CH_2_Cl_2_ (50 mL) were placed in the flask and stirred under inert atmosphere. After 15 min of stirring at room temperature, diisopropyl azodicarboxylate (DIAD, 1.08 mL, 5.5 mmol) was added in small portions (0.15–0.2 mL) into the reaction mixture. The obtained orange color of the solution was allowed to fade prior to adding the next portion DIAD. The solution was left to react under stirring in inert atmosphere at room temperature for 12 h. Then, the reaction mixture was precipitated in diethyl ether (Et_2_O) (1 L) under vigorous stirring in an ice bath for 45 min. The white precipitate was separated from the solvent via suction filtration. The collected product was washed several times with Et_2_O. To achieve high conversion, the reaction was repeated by using the product as a starting reagent. The obtained PE-b-PEG phthalimide (**2**) was precipitated. To improve its purity, it was dissolved in small amount of CH_2_Cl_2_, precipitated in Et_2_O twice and dried under vacuum for 3.5 h. Yield = 91.6 %. FT-IR (cm^−1^) = 3586, 2917, 2849, 1789, 1734, 1640, 1462, 1348, 1324, 1295, 1248, 1187, 1095, 1036, 995, 948, 877, 848, 808, 706, 669, 518. ^1^H NMR (500 MHz, CHCl_3_-d, δ ppm) = 0.88 (t, J = 6.94 Hz, 3 H), 1.25 (s, 54 H), 1.57 (quin, J = 6.94 Hz, 2 H), 3.44 (t, J = 6.94 Hz, 2 H), 3.47–3.52 (m, 1H), 3.53–3.71 (m, 180 H), 3.74–3.81 (m, 1 H), 3.82–3.90 (m, 2 H), 4.34–4.42 (m, 2 H), 7.72–7.87 (m, 4 H).

Synthesis of aminooxy PE-b-PEG **(3)**: A 100 mL flame-dried two neck round flask equipped with stirrer bar was charged with anhydrous CH_2_Cl_2_ (10 mL) under nitrogen. Then, PE-b-PEG phthalimide (**2**) (1.0675 g, 0.446 mmoL) was added and dissolved at room temperature. After 20 min, hydrazine hydrate (58 μL, 0.468 mmoL) was added to the reaction mixture and stirred in inert atmosphere at room temperature for 1 h. The color changed from orange to yellow, and a white precipitate was monitored. Et_2_O (0.5 L) was added to the reaction mixture, placed in an ice bath, under vigorous stirring and left to stir for 2 h. The filtered solid product was rinsed with Et_2_O and dried under vacuum. Yield = 48.2 %. FT-IR (cm^−1^) = 2917, 2849, 1462, 1348, 1324, 1297, 1248, 1095, 1040, 995, 948, 846, 720, 532. ^1^H NMR (500 MHz, CHCl_3_-d, δ ppm) = 0.87 (t, J = 6.94 Hz, 3 H), 1.25 (s, 55 H), 1.57 (quin, J = 7.09 Hz, 2 H), 3.44 (t, J = 6.62 Hz, 2 H), 3.47–3.53 (m, 1H), 3.53–3.76 (m, 190 H), 3.76–3.82 (m, 2 H), 3.84–3.89 (m, 1 H), 4.00–4.04 (m, 1 H), 4.06–4.41 (m, 1 H), 4.06–4.41 (m, 1 H), 4.41–4.50 (m, 1 H)

Functionalization of wt-TMV: To introduce a ketone residue to the virus surface, diazonium salt was prepared from aqueous p-toluenesulfonic acid monohydrate (800 µL, 0.84 M), aqueous sodium nitrite (800 µL, 0.46 M) and p-aminoacetophenone (1.6 mL, 0.15 M in acetonitrile), mixed in a reaction tube and placed in an ice bath. The color of the solution was converted to yellow while stirring for 1 h. Then, virus solution (1033 µL, 4.84 mg/mL) was prepared separately by mixing wt-TMV (500 µL, 10 mg/mL) in 10 mM SPP buffer (pH 7.4) with borate buffer (533 µL, 68 mM) containing 100 mM NaCl (pH 9). 160 µL (11.97 µmoL) diazonium salt solution were mixed with the obtained virus solution and kept in a water bath in the fridge at 4°C for 2 h. The color of the solution turned from yellow to dark brown. The reaction solution was purified using a PD MiniTrap G-25 column with an exclusion limit of approximately M_r_ = 5000 applying a gravity protocol. The column was eluted with 100 mM aqueous potassium phosphate buffer (pH 6). After the purification step, 2 mL containing 2.1 mg/mL TMV-ketone (**4**) were obtained. The synthesis of the product (**4**) was followed by SDS-PAGE analysis.

Synthesis of h-TMV (**4)**: Aminooxy PE-b-PEG (**3**) (0.1614 g, 71.25 μmoL), the TMV-ketone (**4**) (2 mL, 2.1 mg/mL) and 2 mL 100 mM potassium phosphate buffer (pH 6) were mixed in a reaction tube. The mixture was gently rotated overnight at room temperature and purified with a PD MiniTrap G-25 column. Further purification was obtained by size exclusion chromatography at a flow rate of 0.5 mL min^−1^ using a 16/60 Sephacryl S-500 column and 50 mM SPP containing 150 mM NaCl as running buffer. A Pharmacia Biotech LCC-501 PLUS FPLC system with automatic fraction collector was used at room temperature. Fractions exhibiting 1.19 absorbance ratio of 260 nm and 280 nm, indicative for virus particles composed of 95 w/w % proteins and 5 w/w % RNA, were pooled and precipitated by ultracentrifugation at 34 000 rpm at 4°C in a 45 Ti rotor for 2 h. The functionalized virus particles were then resuspended in ultrapure water. The success of the reaction was proved by SDS-PAGE. The height and integrity of the hydrophobized viruses (**5**) was checked by AFM. In a control experiment, wt-TMV (10 µL, 10 mg/mL), borate buffer (10.7 µL, 68 mM) containing 100 mM NaCl (pH 9), potassium phosphate buffer (160 µL, 100 mM, pH 6) and aminooxy PE-b-PEG **(3)** (3.3 mg, 1.46 μmoL) were mixed and rotated overnight at room temperature. The reaction mixture was purified and characterized according to the procedure used for product (**5**).

Sodiumdodecylsulfate-polyacrylamide-gel electrophoresis (SDS-PAGE): Standard procedures according to Green et al. were applied [32]. Samples containing between 0.2–1 µg protein were heated for 5 min at 95 °C in sample buffer and resolved in 15 % discontinuous SDS-PA gels. Fixed gels were stained using a silver staining procedure.

ZnS mineralization: Silicon substrates immobilized with the corresponding virus types were fixed in a holder and mounted perpendicular in the center of a vessel. There, the viruses were incubated in aqueous ZnCl_2_ precursor solution (30 mL, 100 mM, pH 6.2) for 20 min. Then, equimolar aqueous Na_2_S solution (pH 12.5) was added dropwise to the reaction solution applying a peristaltic pump with a constant speed (1 mL min^−1^) under continuous stirring at room temperature. The addition of Na_2_S to the ZnCl_2_ precursor solution caused a drop in the pH to 3.5, and the reaction pH was maintained until the dropwise addition was stopped. Within the addition of the first Na_2_S droplets, a white precipitate was formed in the deposition solution. After finishing the reaction (controlled according to the required reaction time (5 or 30 min)), the substrates were washed thoroughly with ultrapure water and dried under argon stream.

Sample characterization: Atomic force microscopy on a Digital Instruments MultiMode 8 from Bruker with a NanoScope 5 controller operated in tapping mode was used to image the immobilized and the mineralized viruses. Silicon cantilevers and PPP-NCHR-W (Nanosensors) n^+^ doped tips with resistivity 0.01–0.02 Ω cm were used. The virus height of wt-TMV and h-TMV was evaluated with the manufacturer’s software Nanoscope. The height of a single virus was averaged over the whole virus length with the software. The average height of 10 virus particles from each virus type was used to obtain the virus height distribution of the bare virus. The virus height of wt-TMV and h-TMV after treatment with ZnS deposition solution for 5 min, 20 min and 30 min was determined applying the same procedure, taking 5 to 10 virus particles of each virus type for the evaluation and comparing only wt-TMV and h-TMV particles mineralized simultaneously in the same deposition solution. ^1^H-NMR measurements were conducted on Bruker Avance 500 in 500 MHz field and analyzed by ACD/Spectrus Processor. Fourier transform infrared spectroscopy (FT-IR) analysis of the samples was performed at a Bruker FT-IR spectrometer. The samples were investigated in the wavelength range: 500–4000 /cm^−1^ and the spectra were analyzed by ACD/Spectrus Processor.

## 3. Results and Discussion

The wt-TMV particle consists of a genomic RNA helix, embedded into about 2130 identical CP subunits. It has a tube-like structure, which is 300 nm long. The inner diameter of the longitudinal central channel is 4 nm, while the outer nanotube diameter is 18 nm when the capsid is dispersed in solution. It may be flattened to a height of around 14 nm upon immobilization on a hydrophilic silicon substrate surface [12,33]. The viral CPs can be engineered genetically to introduce amino acid residues on the virus surface with certain terminal groups appropriate for bioconjugation reactions. In this study, wt-TMV and a TMV-Cys mutant, presenting a thiol group for coupling on each CP were used to find appropriate synthetic pathways for covalent attachment of organic molecules making the virus surface hydrophobic without destroying the integrity of the capsid, in order to suppress the virus mineralization by inorganic deposits as shown in Scheme 1.

Recently, we reported on selective mineralization of wt-TMV with ZnS at room temperature in aqueous solution and at various reaction pH [10]. ZnCl_2_ and Na_2_S were used as ZnS precursors without the need of any further additives as structure directing or complexing agents. The growth mechanism of the inorganic material on the virus surface was studied comparing the optical properties, band gap (Eg) and particle size of solution-grown ZnS nanoparticles and ZnS nanoparticles mineralized onto the virus template. It was shown that the virus particles played templating function, and heterogeneous nucleation triggered by the virus surface was confirmed. The XRD analysis confirmed formation of ZnS with cubic lattice structure in solution and on the virus surface. Since well-studied, the mineralization of wt-TMV with ZnS applying this reaction was used as a reference in this work and compared to the mineralization behavior of the functionalized virus particles in order to identify, if the change of virus surface hydrophilicity influenced the mineralization process.

### 3.1. Reaction of TMV with Highly Hydrophobic Polymers

A covalent attachment of a highly hydrophobic polymer such as poly(pentafluorostyrene) (PFS) to the virus surface could provide both specific coupling to the virus surface and high surface hydrophobicity. Therefore, first a click reaction of PFS to a TMV-Cys was examined. Our previous study revealed that the reactivity of the para-fluorine function in the PFS with nucleophiles such as sulfides is enhanced by the reduced electron density of the perfluorinated phenyl ring, allowing the use of relatively mild reaction conditions [31]. The reaction is conducted in *N,N*-dimethylacetamide (DMAc) or tetrahydrofuran/ triethylamine (THF/TEA) at room temperature. For the click reaction, TMV-Cys particles were selected, because each of their CPs provides a single surface-accessible thiol group, introduced by genetic modification of the third CP amino acid from serine to cysteine (S3C) [27]. The bioconjugation was conducted according to the reaction pathway presented in Scheme 2A.

Before starting with the coupling reaction, the stability of TMV-Cys in DMAc and in THF with or without TEA was studied at room temperature. To do this, viruses immobilized on a plasma cleaned silicon substrate (Figure 1a) were incubated in DMAc for 30 min. This led to partial loss of the virus integrity especially visible in Figure 1b, where the viruses became roughened and the virus sharp ends are lost. The mean virus height is reduced from 14.5 nm to around 13 nm. Finally, reaction with PFS in DMAc for the same reaction time led to even further disassembly of the viruses, where the virus coat appeared irregular, with an uneven coverage of the RNA with CPs (Figure 1c).

In comparison, the virus structure seemed stable in both THF and THF/TEA for a few hours (Figure 2a,b). This is in contradiction to our previous experiments on the stability of TMV dispersed in THF/H_2_O mixtures, where TMVs appeared to be unstable. The reason for the increased stability in this case might be explained by the immobilization of the viruses on the substrate. Based on the enhanced overall structural stability and the need of an extended reaction time for better educts conversion, the TMV-Cys particles were let to react with PFS in THF/TEA for 2 h at room temperature. The AFM analysis indicated that the capsids stayed intact and retained their integrity after this treatment without change in virus height (Figure 2c). However, subsequent mineralization resulted in deposition of ZnS on the virus surface, and the virus height increased to 16.5 nm (Figure 2d,e). The latter suggests that although stable in this reaction solution, the viruses could not be functionalized with PFS. This may be attributed to the lower reactivity of PFS in THF/TEA compared to that in DMAc, which not only requires longer reaction time but usually also higher temperatures not suitable for TMV.

### 3.2. TMV Hydrophobization with Aliphatic Polyethers

As established in the previous section, the highly hydrophobic environment and the aggressive reaction solvent necessary for dissolving the polymer caused strain on the virus structure and subsequent disintegration. As an alternative, so-called thermo-responsive polymers e.g., poly (propylene glycol)methacrylate can be applied [34]. This polymer can change its hydrophobicity with temperature. It is water-soluble at temperatures below ~10 °C and neutral pH, and becomes insoluble at higher temperatures due to conformational changes inducing hydrophobization. This would allow a chemical coupling of the polymer at 4 °C in its hydrophilic form to the virus surface with expectation that the virus will retain its stability at these conditions. Then, a mineralization test at room temperature or at elevated temperatures, when the polymer becomes hydrophobic, will be conducted to examine the obtained hydrophobicity. Due to the higher reactivity of the acrylate group over the methacrylate group as a Michael addition donor, poly(propylene glycol)acrylate (PPGA) (*M_n_*= 475 g mol^−1^) was chosen for our experiments (Scheme 2B).

To determine the coupling efficiency and specificity, TMV-Cys and wt-TMV were incubated at 4 °C where PPGA is completely soluble (see Appendix A) for 72 h, with PPGA added at different ratios with regard to the number of CPs. SDS-PAGE analysis of the CPs revealed for a ratio of 1000:1 PPGA:CP-Cys only a slight smear above the CP band, which increased substantially at a ratio of 3300:1, but not significantly more at a further excess of PPGA:CP-Cys of 10,000:1 (see Figure 3). The coupling of PPGA to CP-Cys did not result in a defined band due to the polydispersity of PPGA. The absence of any smear above wt-CPs of wt-TMV particles treated in parallel shows on the one hand that the reaction was specific for TMV-Cys, and on the other hand that the TMV structure stayed intact during the reaction, as during TMV particle disassembly an otherwise hidden cysteine residue would become accessible to the reaction with PPGA.

As PPGA was used in high excess, residual unreacted polymer was removed prior to further investigations by a combined ultracentrifugation/fast protein liquid chromatography (UC/FPLC) approach. All purification steps were performed below 5 °C to prevent switching of PPGA into the hydrophobic state. The first ultracentrifugation step was necessary to get rid of the bulk amount of PPGA, which may result in a broadening of the peaks due to overloading size-exclusion chromatography. About 25 % of the TMV initially used for coupling could be obtained in pure form. Figure 4a,b show AFM images of PPGA-functionalized TMV-Cys before and after purification, where residual excess PPGA appears as large spots around the TMV rods in the first image and had disappeared in samples after the cleaning procedure.

After having shown that the thermo-responsive PPGA was successfully coupled to TMV-Cys, the degree of hydrophobicity achieved with PPGA was investigated by subjecting the modified nanotubes to mineralization reaction with ZnS [10]. PPGA-functionalized TMV-Cys viruses were immobilized on a plasma cleaned silicon substrate, which was placed perpendicularly in ZnCl_2_ aqueous solution in order to avoid attachment of big ZnS agglomerates formed in solution on the substrate surface. The mineralization reaction was allowed to proceed for 5 min, which based on previous experiments is enough to observe differences in the mineralization behavior between untreated TMV-Cys and hydrophobized particles, respectively. The amplitude AFM image in Figure 4c clearly shows that the PPGA-functionalized TMV-Cys surface is covered with small ZnS nanoparticles, and the virus height calculated from the corresponding AFM height images was increased by around 1.6 nm. This leads to the conclusion that, although successfully attached to the virus surface preserving the capsid structure, the thermo-responsive PPGA residues could not provide sufficient hydrophobicity to fully suppress virus mineralization.

### 3.3. TMV Hydrophobization with Block-Co-Polymers

Covalent attachment of PPGA to the virus CPs was possible without affecting the overall capsid structure, however, the obtained viruses were not hydrophobic enough to suppress mineralization. A possible reason for this could be the length of the PPGA molecules, and, as obvious from the SDS-PAGE analysis presented in Figure 3, only part of the viral CPs was covalently coupled with PPGA. Another possible reason may be connected to the surface charge of the functionalized viruses. The overall surface charge of organic templates (in our case functionalized TMV) plays an important role for mineralization in solution. The presence of even partial charges like from the free electron pairs of the oxygen in the PPGA polyether can induce mineralization of the PPGA-functionalized viruses. Therefore in the next step, a synthetic pathway developed by Schlick et al [35]. was used. In their work, the TMV capsid exterior was covered with polyethylene glycol (PEG) to make TMV soluble in various hydrophobic organic solvents and to increase its thermal stability. Although PEG introduced sufficient hydrophobicity on the virus surface to transfer the PEG-TMV conjugate from aqueous solution into organic solvents, the presence of oxygen in the PEG chains would support mineralization of the virus, similar to the case of TMV-Cys-PPGA. Therefore, in order to introduce sufficient hydrophobicity without affecting the integrity of the virus structure, we covalently coupled the block-co-polymer (PE)_11_-(PEG)_34_-OH (M_w_ 2250 g/moL) to the wt-TMV surface. The non-polar polyethylene (PE) part is expected to ensure sufficient hydrophobicity of the virus surface and to suppress the mineralization of the organic template. The PEG chain, compatible with the virus, should separate the virus from the hydrophobic PE chain. Additionally, the PEG spacer unit is responsible to attach the block-co-polymer to the virus CP [36]. To do this, we used the already reported electrophilic substitution reaction with diazonium salts at the *ortho* position of the phenyl ring of the tyrosine residue (Y139) in TMV CPs [35]. The polymer coupling was done in a three-step procedure as can be seen in Scheme 3.

To this end, the block-co-polymer **1** converted into the alkoxyamine **3** was reacted with the ketone-functionalized wt-TMV **4** to form the ketoxime **5**. In step I, the terminal hydroxyl group of the block-co-polymer **1** was activated with N-hydroxyphtalimide. As a good leaving group, N-hydroxyphtalimide facilitated the subsequent reaction with hydrazine resulting in the formation of alkoxyamine **3**. The success of the reactions was verified by ^1^H-NMR combined with FT-IR spectroscopy [37]. The appearance of protons with a chemical shift in the low field region around 7.8 ppm and an IR signal in 1730–1790 cm^−1^ corresponding to aromatic and carbonyl residues by product **2** (Figure 5) and their disappearance by product **3** confirmed the attachment and the subsequent replacement of N-hydroxyphtalimide from the polymer.

In step II, the virus CPs were functionalized and a terminal ketone group was introduced, which was done through the accessible tyrosine groups of CPs. It was assumed that only one of the four tyrosine residues, present in each TMV CP, should be modified since the other three are hidden between the CPs after assembly to the helical TMV particle structure [35]. The ketone group was included by treatment of TMV with diazonium salt, prepared prior to use from sodium nitrite and p-aminoacetophenone. The virus functionalization was performed at 4 °C, pH 9 for 2 h. PD MiniTrap G-25 columns were used to separate the obtained TMV-ketone **4** from the added in surplus diazonium salt. SDS-PAGE analysis confirmed the CP modification (see Appendix A) while AFM examination showed preserved virus particles. The hydrophobization of wt-TMV (Scheme 3, step III) was done by reaction of 250 eq. aminooxy PE-b-PEG **3** with TMV-ketone **4** in potassium phosphate buffer (pH 6) at room temperature for 15 h. The hydrophobized TMVs (h-TMV) were purified through PD MiniTrap G-25 columns. Then, the h-TMVs were further separated from the not reacted polymer residue via FPLC and concentrated via UC. The shift of the CP band in the SDS-PA gel (Appendix A) to one with higher molecular weight clearly indicated the successful attachment of polymer to the virus CPs compared to the control experiment, where the aminooxy PE-b-PEG **3** was reacted with a not functionalized wt-TMV and no change in the position of wt-TMV CPs was observed. The blurring of the band corresponding to the CPs of h-TMV can be explained by the polydispersity of the block-co-polymer. The AFM height images in Figure 6 confirmed that the h-TMV particles remained intact. The AFM cross-sections have shown that compared to wt-TMV (Figure 6a,c), the polymer attachment on the virus surface led to an increase in virus height of about 3 nm on hydrophilic (O_2_-plasma treated) (Figure 6b) but also on hydrophobic (not plasma treated) (Figure 6d) silicon substrates.

As stated in previous reports, very polar and protic solvents lead to precipitation of PEG functionalized TMV [36]. Therefore, the stability of the h-TMV in the precursor solution was examined before starting with the mineralization test with ZnS. A silicon substrate with immobilized h-TMV was placed in ZnCl_2_ aqueous solution. After 30 min stirring at room temperature, the viruses could still be detected as intact particles via AFM (see Appendix Aa). Then, a substrate with h-TMV and one substrate with immobilized wt-TMV, used as a reference, were mineralized simultaneously with ZnS for different reaction times. In Figure 7 are shown the corresponding AFM height, amplitude and phase images of wt-TMV and h-TMV after 5 and 30 min treatment. It can be seen that after 5 min reaction time, the surface of wt-TMV became rough due to attachment of ZnS particles, while the surface of h-TMV remained smooth. This is especially visible comparing their AFM amplitude and phase images. Increase of the reaction time to 30 min maintained the trend. The amount of the ZnS particles on wt-TMV surface was increased, which is well detectible also on the AFM height image. Opposed to this, no specific ZnS deposition on h-TMV was observed.

The observation of a selective deposition of ZnS nanoparticles only on the surface of wt-TMV, but not on the surface of the hydrophobized counterpart was further supported by evaluation and comparison of the virus height of wt-TMV and h-TMV treated with ZnS deposition solution for a certain reaction time (Figure 8). Since small fluctuations in the mineralization conditions might cause deviations in the deposition rate and hence might influence the resulting virus height, for better assessment, the virus heights of wt-TMV and h-TMV mineralized in the same deposition solution and within the same experiment were compared. The chart diagram in Figure 8 shows that in comparison to the hydrophobized viruses, whose surface remain smooth with the reaction time, the wt-TMV particles increase their height linearly with the mineralization time. The latter is a result of selective deposition of ZnS on the virus surface with a constant rate, which is in a good agreement with our previous observations for mineralization of wt-TMV with ZnS applying this reaction [10].

## 4. Conclusions

In this work, we have demonstrated that the surface hydrophilicity of TMV particles can be manipulated via covalent attachment of polymer molecules. Experiments with highly hydrophobic PFS, the thermo-responsive PPGA, and the block-copolymer PE-b-PEG led to the conclusion that the success of the reaction depends on the polymer features. The use of perfluorinated polymers showed that highly hydrophobic polymers are not appropriate candidates for virus functionalization. Although the virus particles displayed significant stability enhancement in organic solvents when immobilized on solid supports, the reaction failed due to the need of either aggressive organic solvent (DMAc) or elevated reaction temperature both not compatible with the virus. In contrast, the thermo-responsive PPGA polyether, insoluble in water above 8 °C, was successfully attached to the TMV-Cys surface, and also PE-b-PEG was successfully bonded to wt-TMV CPs. As a result, stable and intact functionalized particles were synthesized. The covalent coupling to virus CPs was confirmed via SDS-PAGE analysis. While PPGA only partially coupled to TMV-Cys CPs, complete coupling efficiency was reached with PE-b-PEG and wt-TMV CPs. The height of the hydrophobized with PE-b-PEG virus (h-TMV), detected by AFM, showed a significant increase by around 2.5 nm. The degree of virus hydrophobicity, achieved with both polymers, was examined by mineralization with ZnS. The incomplete coverage of the virus surface with PPGA and the presence of partial charges in the polymer chains could not suppress virus mineralization. However, combination of a virus-compatible PEG block and a pure hydrophobic PE block in PE-b-PEG together with bioconjugation of the polymer to most CPs in TMV led to the formation of intact h-TMV, fully covered with polymer molecules exposing their hydrophobic parts outwards, which completely hindered subsequent mineralization of the virus surface. This synthetic approach could be used for the synthesis of Janus-type TMV particles with a hydrophobic and a hydrophilic part in future experiments, to study the mineralization behavior of TMV mutants within a single artificial TMV particle with an intrinsic hydrophobic reference. Furthermore, the self-organization of pure and mineralized Janus-type TMV particles in complex highly organized structures as well as their potential integration in interfacial catalysis and as drug cargo is envisioned.

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
