# Peer review of "Hydrophobization of Tobacco Mosaic Virus to Control the Mineralization of Organic Templates"

_nanomaterials, 2019, doi:10.3390/nano9050800_

Round 1

Reviewer 1 Report

Hydrophobization of tobacco mosaic virus to control  the mineralization of organic templates

Petia Atanasova , Vladimir Atanasov, Lisa Wittum, Alexander Southan, Eunjin Choi, Christina Wege, Jochen Kerres, Sabine Eiben and Joachim Bill

Herewith I am submitting my reviewer comments for the above-mentioned manuscript which is under consideration to be published in Nanomaterials. The article is about functionalizing TMV to change hydrophobicity. The resulting material functions as template for synthesis.The authors investigated several different polymers for this modification: perfluorinated (poly(pentafluorostyrene) (PFS)), the thermo-responsive poly(propylene glycol) acrylate (PPGA), and the block-copolymer polyethylene-block poly(ethylene glycol) were examined. The use of TMV as a template is already a rather established process and TMV has been utilized for such applications in multiple studies. Also functionalizing the TMV surface for such a purpose has already been shown. Their main argument is that many of these methods have issues with virus stability. Another concern is that the reaction can be controlled to end at a desired point. Generally, the language is of good quality.

However, there were a few things that I didn't understand contant wise.

 Why would one need the material that was made from TMV templates (I do see the value of this and I am sure this could be cool for something but it would be nice to have a bit better idea what that something is. Maybe you can elaborate a bit more on that)

Most data in the manuscript is from AFM height images. However, there is not much quantification or statistics.I would at least measure multiple heights and make some error bars. The height differences are not very big (and the height overall is also a bit lower than I would have expected at first place).

 Line 275: “The mean virus height is reduced from 14.5 nm to around 13 nm” I thought TMV are usually 18 nm in height? Is that something particular about that strain/method?

I initially understood that  the authors evaluated their products by comparing their ability to mineralize ZnS. But in the end all data I saw about this is in Fig 2 and to my opinion they all look kind of the same. So I do not understand what they are concluding from that picture.

Here are also small text-related comments. Line 133: there is something grammatically wrong in the sentence starting with: “The 134 solution was filtered through 0.2 µm filter”

Line 176: “stirring and let to stir for 2 hours.” Should be “stirring and left to stir for 2 hours.”

Author Response

Response to Reviewer 1 Comments

Point 1: Herewith I am submitting my reviewer comments for the above-mentioned manuscript which is under consideration to be published in Nanomaterials. The article is about functionalizing TMV to change hydrophobicity. The resulting material functions as template for synthesis.The authors investigated several different polymers for this modification: perfluorinated (poly(pentafluorostyrene) (PFS)), the thermo-responsive poly(propylene glycol) acrylate (PPGA), and the block-copolymer polyethylene-block poly(ethylene glycol) were examined. The use of TMV as a template is already a rather established process and TMV has been utilized for such applications in multiple studies. Also functionalizing the TMV surface for such a purpose has already been shown. Their main argument is that many of these methods have issues with virus stability. Another concern is that the reaction can be controlled to end at a desired point. Generally, the language is of good quality.

Response 1: We agree with the reviewer that TMV is a very well-established template, which was also already discussed in the introduction part of the manuscript. What makes our work specific is that the hydrophobization is done through a covalent attachment of organic molecules, which makes the surface enough hydrophobic to suppress subsequent mineralization. Moreover, the covalent attachment allows local modification in contrast to the known and reviewed in the introduction examples of not specific hydrophobization methods based on only electrostatic interactions. This method is expected to ensure a local functionalization of only a specific part of the virus particle. In addition to that, attachment of such hydrophobic molecule, which is in the same time compatible with the organic template is not a trivial work and it should fulfil certain requirements as discussed in the manuscript.

Point 2: Why would one need the material that was made from TMV templates (I do see the value of this and I am sure this could be cool for something but it would be nice to have a bit better idea what that something is. Maybe you can elaborate a bit more on that)

Response 2: Our main idea behind searching for an appropriate method for covalent attachment of organic molecules, which makes the TMV surface hydrophobic is the synthesis of Janus-type TMV particles, consisting of two domains one hydrophilic and one hydrophobic. To the best of our knowledge there are no such viruses synthesized, yet. The amphiphilicity of the classical Janus particles provides unique chemical and physical properties, which are not accessible for their homogeneous counterparts. The presence of a hydrophobic domain gives the opportunity to use such Janus-type TMV particles to study the mineralization behaviour (the nucleation and growth) of TMV mutants within a single artificial TMV particle with an intrinsic hydrophobic reference. An additional perspective is envisioned in a self-organization of pure and mineralized Janus-type TMV particles in complex highly organized structures as well as their potential integration in interfacial catalysis and as drug cargo.

All these aspects are discussed in the introduction part and also mentioned as a vision for a future work in the conclusion of the manuscript.

Point 3: Most data in the manuscript is from AFM height images. However, there is not much quantification or statistics.I would at least measure multiple heights and make some error bars. The height differences are not very big (and the height overall is also a bit lower than I would have expected at first place).

Response 3: According to the reviewer’s recommendations, the virus height distribution is presented in Figure 8, and additional text explaining this figure is included in the text. Although the height differences are not big, the trend is clear – wt‑TMV increases in height with the reaction time due to the selective deposition of ZnS on the virus surface with a linear growth rate, while the hydrophobized viruses do not show specific mineralization on their surface (Figure 7, AFM amplitude and phase images).

Point 4: Line 275: “The mean virus height is reduced from 14.5 nm to around 13 nm” I thought TMV are usually 18 nm in height? Is that something particular about that strain/method?

Response 4: TMV has diameter of 18 nm when it is dispersed in solution. However, it is flattened upon immobilization on hydrophilic substrate surfaces. The resulting virus height can vary in the range 14 – 15 nm depending on how much hydrophilic is the substrate surface. We regularly immobilize wt‑TMV and different TMV mutants in our lab, and in all cases such height reduction is observed by their immobilization on glass or silicon substrates. Reference papers are given in the manuscript (ref. 12 and 33).

In this particular case, the virus height is reduced from 14.5 nm to 13 nm due to a partial particle disassembly caused by the presence of N,N‑dimethylacetamide organic solvent.

Point 5: I initially understood that  the authors evaluated their products by comparing their ability to mineralize ZnS. But in the end all data I saw about this is in Fig 2 and to my opinion they all look kind of the same. So I do not understand what they are concluding from that picture.

Response 5: Yes, we have evaluated the achieved virus hydrophobicity by testing the mineralization ability of the modified virus particles. The main result, showing a successful covalent attachment of the block-co-polymer to TMV, forming stable particles which are enough hydrophobic and do not allow mineralization on their surface is shown on Figure 7.

            In Figure 2 are presented the results of our investigation on the stability of TMV-Cys virus particles in tetrahydrofuran (THF), used as a solvent for the reaction with the highly hydrophobic polymer poly (pentafluorostyrene) (PFS). The viruses showed retained integrity after this treatment with THF without change in virus height. However, the functionalization with PFS was not successful, which was confirmed in a subsequent mineralization step and the results are shown on the AFM images in Figure 2.

Point 6: Here are also small text-related comments. Line 133: there is something grammatically wrong in the sentence starting with: “The 134 solution was filtered through 0.2 µm filter”

Response 6: The number “134” should come from the line number given by the text formatting. The sentence is “The solution was filtered through 0.2 µm filter,….

Point 7: Line 176: “stirring and let to stir for 2 hours.” Should be “stirring and left to stir for 2 hours.”

Response 7: The mistake was corrected.

Reviewer 2 Report

The subject matter of this manuscript is relevant to the Nanomaterials aims and scope. The title does adequately describe the contents of this review and the abstract is informative enough. The submitted manuscript in the present form is marginally suitable for publication in the Nanomaterials and it will be further improved after some additions/corrections.

In the first part of this work, authors described very adequately the modification processes of TMV and the respective characterization of the modified substrates.  This part of work is very informative and the results are verified by the experiments.

In the second part of this work, the modified substrates are evaluated with respect to ZnS mineralization. This evaluation is based in the comparison of AFM photos in the presence and absence of modified substrates and is very coarse. The addition of sodium sulfide in zinc chloride solutions under non-constant solution pH is not suitable technique for the evaluation of modified surfaces with respect to their mineralization ability. The pH solution during addition of sodium sulfide becomes more alkaline, thus the formation of insoluble zinc hydroxide is possible. Also, authors did not give any information about the formed solids. Also, is not clear that these solids, on the modified surfaces or in the solution bulk initially are formed. They claimed that the formed solids are zinc sulfide without any evidence. Probably, the use of pH-static method will provide more accurate evaluation of modified surface with respect to zinc sulfide mineralization and improve further the submitted manuscript.

Some additional comments:

1)      Authors must add the data for the reference with number 10 at line 511.

2)      In the reference 32 at line 569, the last number of publication is not given and the character ‘&’ is appeared.

3)      The references 38, 39 and 40 are not noted in the text.

4)      Also, data of the reference 40 are not appeared at lines 586 and 587.

Author Response

Response to Reviewer 2 Comments

Point 1: In the first part of this work, authors described very adequately the modification processes of TMV and the respective characterization of the modified substrates.  This part of work is very informative and the results are verified by the experiments.  

Response 1: Thank you very much for the positive judgment.

Point 2: In the second part of this work, the modified substrates are evaluated with respect to ZnS mineralization. This evaluation is based in the comparison of AFM photos in the presence and absence of modified substrates and is very coarse. The addition of sodium sulfide in zinc chloride solutions under non-constant solution pH is not suitable technique for the evaluation of modified surfaces with respect to their mineralization ability. The pH solution during addition of sodium sulfide becomes more alkaline, thus the formation of insoluble zinc hydroxide is possible.

Probably, the use of pH-static method will provide more accurate evaluation of modified surface with respect to zinc sulfide mineralization and improve further the submitted manuscript.

Response 2: Thank you very much for this important point, and hopefully our explanation will give complete answer of your questions. We have used this very simple reaction with purpose, because of the following reasons: a) no additives are needed for the formation of ZnS, and this is very important, because additives might contribute to the organic-inorganic interactions and make difficult to study pure interaction between the virus and the inorganic material, b) the reaction is conducted at RT and in aqueous solution, which are very advantageous for the viruses, c) the reaction pH is suitable for TMV (the viruses are stable at pH between 2 and 9) and d) the mineralization of wt‑TMV with ZnS applying this reaction is very well studied and characterized in our group. The results are already published in a separate paper (reference 10 in the revised version of the manuscript: Atanasova, P., Kim, I., Chen, B., Eiben, S., Bill, J. Controllable Virus-Directed Synthesis of Nanostructured Hybrids Induced by Organic/Inorganic Interactions. Adv. Biosys. 2017, 1700106.)

The reaction is conducted at constant pH. The Na2S precursor solution has pH above 12 (at this value the viruses would be completely disassembled), while the pH of ZnCl2 is 6.1-6.5 depending of the batch (therefore the whole set of experiments were conducted with the same precursor batch). When Na2S is added dropwise to ZnCl2 solution with the virus immobilized substrates, the pH drops immediately to pretty low but certain pH in the range (3.5-4.5 – again depending on the precursor batch), and it is maintained constant until the addition of Na2S is stopped.

The mineralization ability of wt-TMV is comprehensively studied and the results are presented in the above mentioned paper. A pH-controller was used to fix the pH and study the ZnS mineralization on TMV at different pH.

To improve the manuscript content according to the reviewer’s comments, additional text is included in the revised version giving further information for the reaction.

Point 3: Also, authors did not give any information about the formed solids. Also, is not clear that these solids, on the modified surfaces or in the solution bulk initially are formed. They claimed that the formed solids are zinc sulfide without any evidence.

Response 2: The amount of the deposited inorganic material on the virus surface in this study (applying only short deposition time) is not sufficient for XRD measurements. The latter together with the very small ZnS particle size (~ 1-2 nm, confirmed from the band gap, measured via UV-Vis) did not allow to observe a XRD pattern. However, our investigations on the composition and crystallinity of the deposited material were already conducted on wt‑TMV/ZnS hybrid films prepared after 20 deposition cycles (published in reference 10). As can be seen in Fig. 5 in this paper, “the XRD pattern of pure ZnS particles precipitated in the solution shows broad peaks (Figure 5a) which appear at 2θ ≈ 28.7°, 2θ ≈ 48.1°, and 2θ ≈ 56.3° and could be assigned to the (111), (220), and (311) planes, respectively of ZnS cubic lattice structure. The latter corresponds to zinc blend also referred to as the sphalerite polytype. The XRD patterns of TMV/ZnS films formed at various pH have shown similar peak profiles (Figure 5b) with a relative increase of the peak broadening due to formation of small crystals.” Therefore, in our work here we only refer to this paper, where the characterization of the wt-TMV/ZnS hybrid is already presented.

The mineralization mechanism of the reaction was also presented in ref. 10 and the information was mentioned in our manuscript as follows (in the revised version: line 255-256): “ZnCl2 and Na2S were used as ZnS precursors without the need of any further additives as structure directing or complexing agents. The virus particles played templating function, and heterogeneous nucleation triggered by the virus surface was confirmed.”

In accordance with the reviewer’s comments, we have included additional information in the revised version for the mineralized inorganic material.

Point 4: Some additional comments:

1)      Authors must add the data for the reference with number 10 at line 511.

2)      In the reference 32 at line 569, the last number of publication is not given and the character ‘&’ is appeared.

3)      The references 38, 39 and 40 are not noted in the text.

4)      Also, data of the reference 40 are not appeared at lines 586 and 587.

Response 4: The requirements in points 1), 3) and 4) are fulfilled. These mistakes should have appeared through the creation of automatically given numbers for the lines by the text formatting.

The reference from point 2) has only one page and the symbol & is removed.

Reviewer 3 Report

In this manuscript, the authors address very interesting topic and presented results of the surface properties modification of tabacco mosaic virus by polymer coupling in order to change its mineralization. The paper is written very well. I have no objections to the methodology, results presentation and conclusions.

Author Response

Response to Reviewer 3 Comments

In this manuscript, the authors address very interesting topic and presented results of the surface properties modification of tabacco mosaic virus by polymer coupling in order to change its mineralization. The paper is written very well. I have no objections to the methodology, results presentation and conclusions.

Response 1: Thank you very much for your very positive report!

Reviewer 4 Report

This study describes a functionalization strategy to improve the hydrophobicity of tobacco mosaic virus (TMV)  surfaces as a means to restrict mineralization.  The authors envision this hydrophobization approach to have future applications in synthesizing Janus-type TMV particles with a hydrophobic and a hydrophilic part. The manuscript is well written with clear objectives and methodologies. The experiments are carefully designed and executed. Though its an incremental work, the chemistry proposed by the authors for hydrophobization of TMV can be utilized for wider applications.  There are no major issues in this manuscript. However, it is recommended that in Fig. 7, the authors provide the height image and distribution of particle size before and after mineralization for wt-TMV vs h-TMV respectively.

Author Response

Response to Reviewer 4 Comments

This study describes a functionalization strategy to improve the hydrophobicity of tobacco mosaic virus (TMV) surfaces as a means to restrict mineralization.  The authors envision this hydrophobization approach to have future applications in synthesizing Janus-type TMV particles with a hydrophobic and a hydrophilic part. The manuscript is well written with clear objectives and methodologies. The experiments are carefully designed and executed. Though its an incremental work, the chemistry proposed by the authors for hydrophobization of TMV can be utilized for wider applications.  There are no major issues in this manuscript. However,

Point 1: It is recommended that in Fig. 7, the authors provide the height image and distribution of particle size before and after mineralization for wt-TMV vs h-TMV respectively.

Response 1: Thank you very much for the positive feedback. According to the reviewer’s recommendation, Figure 7 is modified and additionally to the amplitude also the corresponding height and phase AFM images are included. The virus height distribution is presented in Figure 8, and additional explanation to this figure is included in the text.

Round 2

Reviewer 1 Report

The comment on line 133: The number just was a typo from copying the text in from the manuscript but that was not what I meant. The sentence still should be revised.